# Generation of 3D Human Models and Animations Using Simple Sketches

Alican Akman*
Koç University

Yusuf Sahillioğlu†
Middle East Technical University

T. Metin Sezgin‡
Koç University

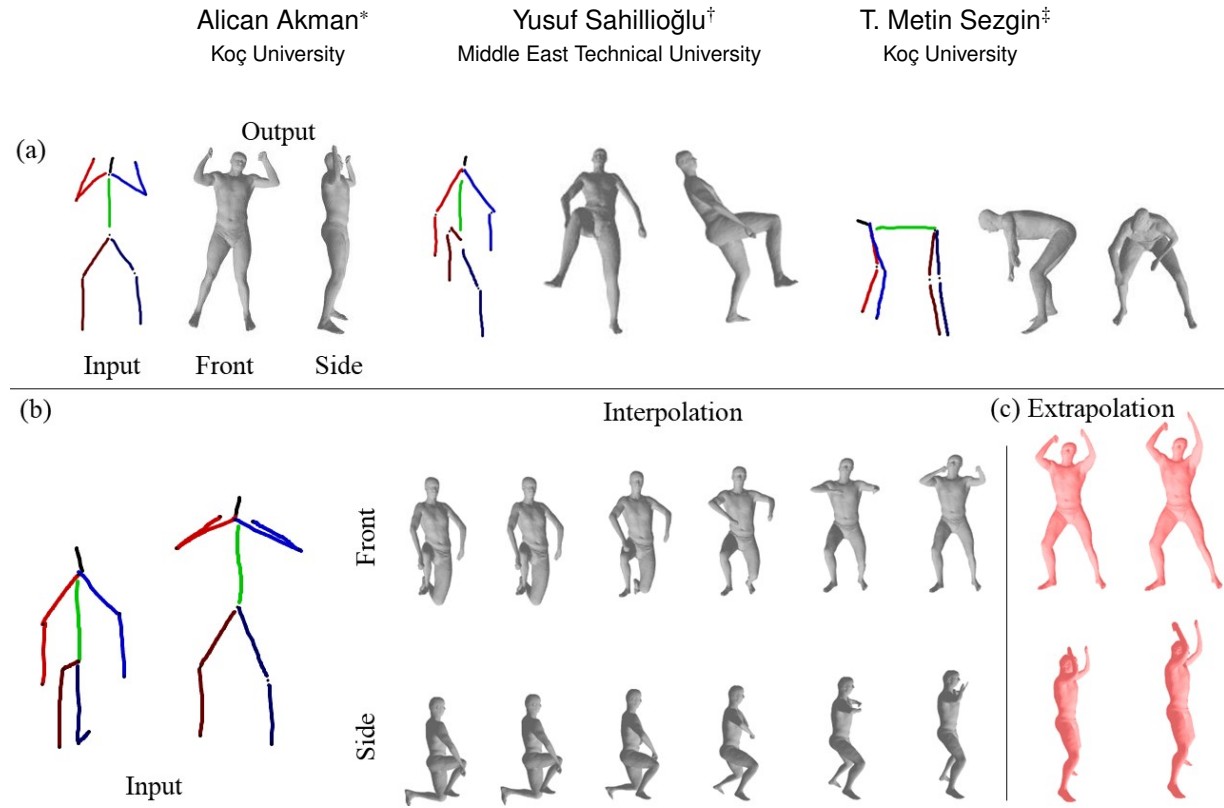

Figure 1: Our framework is capable of performing three tasks. (a) It can generate 3D models from given 2D stick figure sketches. (b) It can generate dynamic 3D models, i.e., animations, between given source and target stick figures. (c) It can further extrapolate the produced 3D model sequence by using the learned interpolation vector.

## ABSTRACT

Generating 3D models from 2D images or sketches is a widely studied important problem in computer graphics. We describe the first method to generate a 3D human model from a single sketched stick figure. In contrast to the existing human modeling techniques, our method requires neither a statistical body shape model nor a rigged 3D character model. We exploit Variational Autoencoders to develop a novel framework capable of transitioning from a simple 2D stick figure sketch, to a corresponding 3D human model. Our network learns the mapping between the input sketch and the output 3D model. Furthermore, our model learns the embedding space around these models. We demonstrate that our network can generate not only 3D models, but also 3D animations through interpolation and extrapolation in the learned embedding space. Extensive experiments show that our model learns to generate reasonable 3D models and animations.

**Index Terms:** Computing methodologies—Neural networks—;——Computing methodologies—Learning latent representations—;

*e-mail: alicanakman4@gmail.com

†e-mail: ys@ceng.metu.edu.tr

‡e-mail: mtsezgin@ku.edu.tr

Computing methodologies—Shape modeling—;—

## 1 INTRODUCTION

3D content is still not as big as image and video data. One of the main reasons of this lack of abundance is the labor going into the creation process. Despite the increasing number of talented artists and automated tools, it is obviously not as simple and quick as hitting a record button on the phone.

3D content is, on the other hand, as important as the image and video data since it is used in many useful pipelines ranging from 3D printing to 3D gaming and filming.

With these considerations in mind, we aim to make the important 3D content creation task simpler and faster. To this end, we train a neural network over 2D stick figure and corresponding 3D model pairs. Utilization of easy-to-sketch and sufficiently-expressive 2D stick figures is a unique feature of our system that makes our system work properly even with a moderate amount of training, e.g., 72 distinct poses of a human model are used. We focus on human models only as they come forward in 3D applications.

Given 2D human stick figure sketches, our algorithm is able to produce visually appealing 3D point cloud models without requiring any other input such as a rigged template model. After an easy and seamless tweaking in the network, the system is also capable of producing dynamic 3D models, i.e., animations, between source and target stick figures.

## 2 RELATED WORK

Thanks to their natural expressiveness power, sketches are common modes for interaction for various graphics applications [30, 39].

The majority of sketch-based 3D human modeling methods deal with re-posing a rigged input model under the guidance of user sketches. [15] performs this action by transforming imaginary lines running down a character's major bone chains, whereas [32] and [16] propose incremental schemes that pose characters one limb at a time. 2D stick figures to pose characters benefit from user annotations [9], specific priors [26], and database queries [8, 45]. Bessmeltsev et al. [5] claim that ambiguity problems of all these methods can be alleviated by contour-based gesture drawing. Deep regression network of [17] utilizes contour drawing to allow face creation in minutes. Another system which takes one or more contour drawings as its input uses deep convolutional neural networks to create variety of 3D shapes [10]. We avoid the need for a rigged model in input specification and merely require a user sketch, which, when fed into our network, produces the 3D model quickly in the specified pose. As the network is trained with the SCAPE models [3], our resulting 3D shape looks like the SCAPE actor, i.e., a 30 year-old fit man.

There also exist sketch-based modeling methods for other specific objects such as hairs [13, 37] and plants [2], as well as general-purpose methods that are not restricted to a particular object class. These generic methods, consequently, may not perform as accurately as their object-specific counterparts for those objects but still demonstrate impressive results. 3D free-form design by the pioneer Teddy model [19] is improved in FiberMesh [29] and SmoothSketch [21] by removing the potential cusps and T-junctions with the addition of features such as topological shape reconstruction and hidden contour completion. The recent SymmSketch system [28] exploits symmetry assumption to generate symmetric 3D free-form models from 2D sketches. In order to increase quality in generating 3D models, [42] focuses piecewise-smooth man-made shapes. Their deep neural network-based system infers a set of parametric surfaces that realize the drawing in 3D. Other solutions to the sketch-based generic 3D model creation problem depend on image guidance [43, 47], snapping one of the predefined primitives to the sketch by fitting its projection to the sketch lines [41], and controlled curvature variation patterns [25].

3D model generation and editing have been extended to 3D scenes as well. Dating back to 1996 [48], this line of works generally index 3D model repositories by their 2D projections from multiple views and retrieve the elements that best match the 2D sketch query [23, 40]. Xu et al. [46] extend this idea further by jointly processing the sketched objects, e.g., while a single computer mouse sketch is not easy to recognize, other input sketches such as computer keyboard may provide useful cues. Sketch-based user interfaces arise in 2D image generation as well [7, 12].

Sketches also arise frequently in shape retrieval applications due to their simplicity and expressiveness. Our focus, the human stick figure sketch, has been used successfully in [35] to retrieve 3D human models from large databases. The prominent example in this domain [11] as well as the convolutional neural network based method [44] report good performance with gesture drawings when it comes to retrieving humans. These three methods, as well as many other sketch-based retrieval methods [24], are in general successful on retrieving non-human models as well.

Although human body types under the same pose can be learned easily with moderate amounts of data through statistical shape modeling [1, 38], this approach requires much greater amounts of input data to learn plausible shape poses under various deformations [18, 33]. In addition to the data issue, this family of methods that are based on statistical analysis of human body shape operate directly on vertex positions, which brings the disadvantage that rotated body parts have a completely different representation. This issue is addressed with various heuristics, most successful of which leads to

the SMPL model [27] that enables 3D human extraction from 2D images [20, 31]. Our learning-based solution requires moderate amount of training data, and also alleviates the rotated part issue by simply populating the input data with 17 other rotated versions of each model.

## 3 OVERVIEW

We have two main objectives: (i) Generating 3D models from a single sketched stick figure, (ii) creating 3D animations between two 3D models, generated from 2D source and target sketches. In addition, we present an application that allows interactive modeling using our algorithm.

Our approach is powered by a Variational Autoencoder network (VAE). We train this network with pairs of 3D and 2D points. The 3D points come from the SCAPE 3D human figure database, while the 2D points are obtained by projecting joint positions of these models on a 2D surface. Hence the correspondence information is preserved. Our neural network model ties the 2D and 3D representations through a latent space, which allows us to generate 3D point clouds from 2D stick figures.

The latent space that ties the 2D and 3D representations also acts as a convenient lower dimensional embedding for interpolation and extrapolation. Given a set of target key frames in the form of 2D drawings, we can map them into the lower dimensional embedding space, and then interpolate between them to obtain a number of via points in the embedding space. These via points can then be mapped to 3D through the network to obtain a smooth 3D animation sequence. Furthermore, extrapolation allows extending the animation sequence beyond the target key frames.

## 4 METHODOLOGY

Our method aims to generate static and dynamic 3D human models from scratch, that is, we require only the 2D input sketch and no other data such as a rigged 3D model waiting to be reposed. To make this possible, we learn a model that maps 2D stick figures to 3D models.

### 4.1 Training Data Generation

The original SCAPE dataset consists of 72 key 3D meshes of a human actor [3]. It also contains point-to-point correspondence information between these distinct model poses. We use a simple algorithm to extend this dataset by rotating the existing figures with different angles. First, we determine the axes and the angles of the rotation with respect to the original coordinate system shown in the wrapped figure. We ignore the rotation

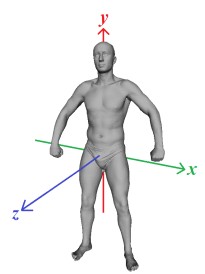

with respect to the *x*-axis, since stick figures are less likely to be drawn from this view. Next, we rotate the models with respect to the *y*-axis and *z*-axis, in a range of -90 degree to 90 at intervals of 30 degrees for the *y*-axis, and -45 to 45 degrees at intervals of 45 degrees for the *z*-axis. In the end of this process, we output 21 models per key model in the SCAPE dataset.

Since our network is trained with (2D joints, 3D model) pairs, we also extract 2D joints from a 3D model in a particular perspective. We designate 11 essential points that alone can describe a 3D human pose. These are the following: forehead, elbows, hands, neck, abdomen, knees and feet. Since the dataset has the point-to-point correspondence information in itself, we select these 3D points in a pilot mesh from the dataset. We project these joints onto a 2D camera plane (*x*-*y* in our case) across our entire dataset to create 2D joint projections. In order to be independent from the coordinate system, we represent these points with relative positions ($\Delta x$, $\Delta y$). We determine a specific order in 2D points forming a sketching path

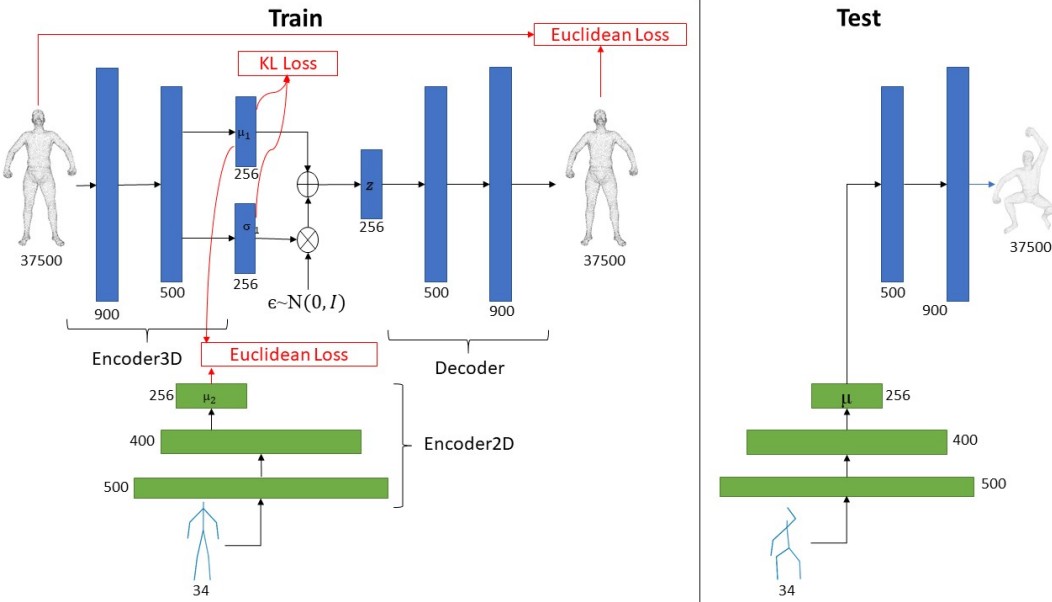

Figure 2: Our neural network architecture. **(a) Train Network:** We train this network with (3D point cloud, 2D points of stick figure) pairs during training time. It consists of a VAE: Encoder3D and Decoder consecutively, and another external encoder: Encoder2D. We use regression loss from the output of Encoder2D to the mean vector of the VAE in addition to standard losses of VAE. **(b) Test Network:** We remove Encoder3D and reparameterization layer from our VAE and use Encoder2D-Decoder as our network in our experiments.

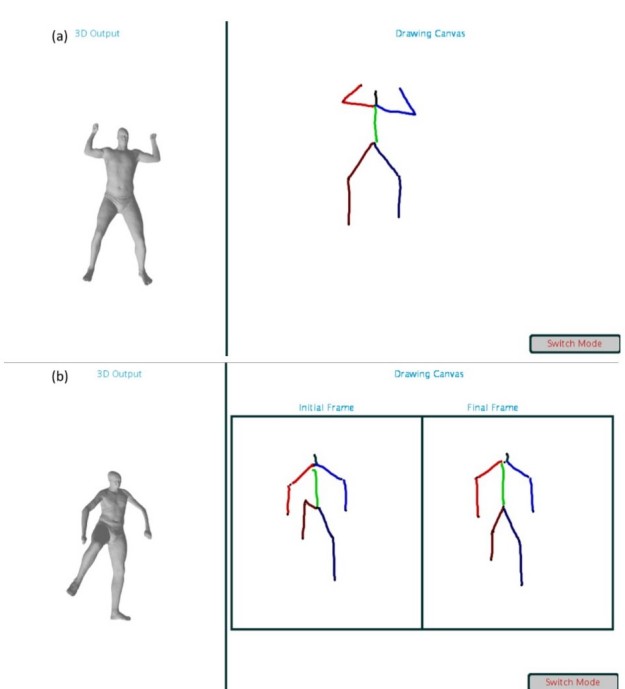

Figure 3: Screenshots from our user interface. (a) 3D model generation mode. (b) 3D animation generation mode.

with 17 points (some joints are visited twice but in reverse direction). This sketching path determines the order of 2D points that form the input vector of our neural network. The input vector format also handles front/back ambiguity while generating 3D models. We set the first point in the sketching path as the origin, $(0,0)$ and then we set the remaining points with respect to their relative position to the preceding point.

## 4.2 Neural Network Architecture

We build upon the work of Girdhar et al. [14] while designing our neural network architecture. Girdhar et al. aim to learn a vector representation that is *generative*, i.e., it can generate 3D models in the form of voxels; and *predictable*, i.e., it can be predicted by a 2D image. We utilize variational autoencoders rather than standard autoencoders to build our neural network as shown in Figure 2. Unlike standard autoencoders, VAEs are generative networks whose latent distribution is regularized during the training in order to be close to a standard Gaussian distribution. This property of VAEs ensures that its latent distribution has a meaningful organization which allows us to generate novel 3D models by sampling in this distribution. In addition to generating novel 3D models, since our framework is capable of learning the vector space around these 3D models, it enables meaningful transitions between them and extrapolations beyond them.

For our training network, we have two encoders and one decoder: Encoder3D, Encoder2D, and Decoder. Encoder3D and Decoder together serve as a VAE. Our VAE takes in a 3D point cloud as input, and reconstructs the same model as the output. While our VAE learns to reconstruct 3D models, it forces latent distribution of the dataset to approximate normal distribution which makes the latent space interpolatable. Meanwhile, we use our Encoder2D to predict latent vectors of corresponding 3D models from 2D points. In order to provide our latent distribution with similarity information between 3D models, we design this partial architecture for our neural network instead of using a VAE which directly generates 3D models from 2D sketches. Thus, our Encoder3D is capable of learning relations between 3D models rather than 2D sketches while creating

a regularized latent distribution. With this method, we aim to explore latent space better and generate more meaningful transitions between 3D models.

**Encoder3D-Decoder VAE Network Architecture:**

Our VAE takes $12500 \times 3$ point cloud of human 3D model as input. Encoder3D contains two fully connected layers and its outputs are a mean vector and a deviation vector. We use ReLu as an activation layer in all the internal layers. There is no activation layer in the output layers. Our Decoder takes the latent vector $z$ as input. It also consists of two fully connected layers with a ReLu activation layer and one fully connected output layer with a *tanh* activation layer. It gives a reconstructed point cloud of the input 3D model as the output.

We train our VAE with the standard KL divergence and reconstruction losses. The total loss for our VAE is given in Equation 1.

$$L_{\text{VAE}} = \alpha \frac{1}{BN} \sum_{i=1}^{B} \sum_{j=1}^{N} (x_{ij} - \hat{x}_{ij})^2 + D_{\text{KL}}(q(z|x)||p(z)) \quad (1)$$

In Equation 1, $\alpha$ is a parameter to balance between the reconstruction loss and KL divergence loss, $B$ is the training batch size, $N$ is the 1D dimension of vectorized point cloud (37500 in our case), $x_{ij}$ is the $j$-th dimension of $i$-th model in the training data batch, $\hat{x}_{ij}$ is the $j$-th dimension of model $i$'s output from our VAE, $z$ is the reparameterized latent vector, $p(z)$ is the prior probability, $q(z|f)$ is the posterior probability, and $D_{\text{KL}}$ is KL divergence.

**Mapping 2D Sketch Points to Latent Vector Space:**

Our Encoder2D learns to predict the latent vectors of 3D models that corresponds to 2D sketches as discussed. It takes $11 \times 2$ points as input to map it into mean vector. It has the same structure with Encoder3D except its input and internal fully connected layers' dimensions. In the test case we use Encoder2D and Decoder as a standard autoencoder. Decoder takes the mean vector output of Encoder2D as its input and generates 3D point cloud as the output.

We train our Encoder2D with mean square loss to regress 256D representation of mean vector given by pre-trained Encoder3D. The loss for our Encoder2D is given in Equation 2.

$$L_{\text{2D}} = \frac{1}{BZ} \sum_{i=1}^{B} \sum_{j=1}^{Z} (\mu_i^1 - \mu_i^2)^2 \quad (2)$$

In Equation 2, $B$ is the training batch size, $Z$ is the dimension of latent space (256 in our case), $\mu_{ij}^1$ is the $j$-th dimension of mean vector produced by Encoder3D to $i$-th model in training batch, and $\mu_{ij}^2$ is the $j$-th dimension of mean vector produced by Encoder2D to $i$-th model in the training batch.

## 4.3  Training Details

We follow a three-stage process to train our network. (i) We train our variational auto encoder independently with the loss function in Equation (1). We run this stage for 300 epochs. (ii) We train our Encoder2D with the loss function in Equation (2) using Encoder3D trained from (i). Specifically, we train our Encoder2D to regress the latent vector produced from the pre-trained Encoder3D for the input 3D model. We run this stage for 300 epochs. (iii) We use both losses jointly to finetune our framework. We run this stage for 200 epochs. It takes about two days to complete whole training session.

For the experiments throughout this paper, we set $\alpha = 10^5$. We set the prior probability over latent variables as a standard normal distribution, $p(z) = \mathcal{N}(z; 0, I)$. We set the learning rate as $10^{-3}$. We use the Adam Optimizer as our optimizer in training.

## 4.4  User Interface

As we have explained in prior sections, our method can be used in a variety of applications in different fields such as character generation and making quick animations. In order to better utilize these applications we propose a user interface with facilitative properties that enables users to perform our method in a better manner (Figure 3). Our user interface acts as an agent that ensures the input-output communication between the user and our neural network. The user can choose whether to generate 3D models from 2D stick figure sketches or create animations between the source and target stick figure sketches. While it takes about one second to process a sketch input for generation of the corresponding 3D model, this time extends approximately to five seconds for producing animation between (interpolation) and beyond (extrapolation) sketch inputs.

Our user interface takes a sketch input from the user via an embedded 2D canvas (right part). The collected sketch is transformed into a map of the joint locations as an input to our neural network by requiring users to sketch in a predetermined path. The user interface then shows the 3D model output produced by the neural network on the embedded 3D canvas (left part). Although our output is a 3D point cloud, for better visualization, our user interface utilizes the mesh information that already exists in the SCAPE dataset. Generated point clouds are consequently combined with this information to display 3D models as surfaces with appropriate rendering mechanisms such as shading and lighting.

Since we trained an abundance of neural networks until achieving the best one with the optimal parameters, our user interface showed two different 3D model outputs coming from two different neural networks for comparisons during the development phase. The interface in Figure 3 belongs to our release mode where only the promoted 3D output is displayed.

We construct our user interface such that it is purified from unnatural interaction tools such as buttons and menus. Generation process starts as soon as the last stroke is drawn without forcing the user to hit the start button. We provide brief information in text that describes the canvas organization. To make the interaction more fluent, we add a simple "scribble" detector to understand the undo operation.

## 5  EXPERIMENTS AND RESULTS

In this section, we first evaluate our framework qualitatively and quantitatively. We evaluate three tasks performed by our framework. (i) Generating 3D models from 2D stick figure sketches. (ii) Generating 3D animations between source and target stick figure sketches. (iii) Performing simple extrapolation beyond stick figures.

## 5.1  Framework Evaluation

**Standard Autoencoder Baseline.** To quantitatively justify our preference on variational autoencoders, we design a standard autoencoder (AE) baseline with similar dimensions and activation functions (Figure 4). We train this network for 300 epochs with Euclidean loss between generated 3D models and ground truth ones. We compare the per-vertex reconstruction loss on validation set consisting held-out 3D models with our VAE network and standard AE network. The results on Table 1 shows that our VAE network outperforms AE network, exploiting latent space more efficiently to enhance the generation quality of novel 3D models.

**Latent Dimensions.** We evaluate different latent dimensions for our VAE framework using per-vertex reconstruction loss on validation set. The results on Table 2 show that using 256 dimensions improves generation quality compared to lower dimensions. Higher dimensions lead to overfitting. We use 256 dimensions for the following experiments.

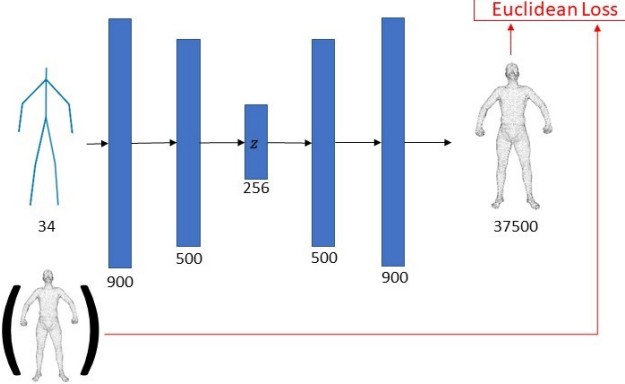

Figure 4: Neural network architecture for standard autoencoder baseline.

Table 1: Per-vertex reconstruction error on validation data with different network architectures. VAE represents our method.

| Method | Z (Latent Dim.) | Mean (x$10^{-3}$) | Std (x$10^{-3}$) |
|--------|-----------------|-------------------|------------------|
| AE     | 256             | 2.996             | 1.568            |
| VAE    | 256             | 2.118             | 2.598            |

Table 2: Per-vertex reconstruction error on validation data with different latent dimensions. 256 is our promoted latent space dimension.

| Method | Z (Latent Dim.) | Mean (x$10^{-3}$) | Std (x$10^{-3}$) |
|--------|-----------------|-------------------|------------------|
| VAE    | 128             | 3.887             | 3.802            |
| VAE    | 256             | 2.118             | 2.598            |
| VAE    | 512             | 10.830            | 6.888            |

## 5.2 Generating 3D models from 2D stick figure sketches

To evaluate the generation ability, we feed 2D stick figure sketches as input to our framework. Our user interface is used for the sketching activity. In Figure 5, we compare generation results for sample sketches with our VAE network and standard AE baseline. These results show that standard AE network generates models with anatomical flaws (collapsed arm in Figure 5) and deficient body orientations. Our VAE network produces compatible models of high quality.

We also compare the generation ability of our framework with a recent study [20] that predicts 3D shape from 2D joints. While our framework outputs 3D point clouds, the method described in [20], tries to fit a statistical body shape model, SMPL [6], on 2D joints. Their learned statistical model is a male who is in a different shape than our male model as shown in the second column of Figure 6. We take the visuals reported in their paper and draw the corresponding stick figures for a fair comparison. Despite being restricted to the reported poses in [20], our method compares favorably. The sitting pose, for instance, which is not quite captured by their statistical model shows in an inaccurate 3D sitting while our 3D model sits successfully (Figure 6 - top row).

We finally compare our 3D human model generation ability to the ground-truth by feeding sketches that resemble 3 of the 72 SCAPE poses used during training. Two of these poses were observed during training at the same orientations as we draw them in the first two rows of Figure 7, yet the last one is drawn with a new orientation that was not observed during training (Figure 7-last row). Consequently, we had to align the generated model with the ground-truth model via ICP alignment [4] for the last row prior to the comparison. We observe almost perfect replications of the SCAPE poses in all cases as depicted by the colored difference maps.

## 5.3 Generation of Dynamic 3D Models - Interpolation

We test the capability of our network to generate 3D animations between source and target stick figures. To accomplish this, our framework takes two stick figure sketches via our user interface. The framework then encodes the stick figures into the embedding space to extract their latent variables. We create a list of latent variables by linearly interpolating between the source and the target latent variables. We feed this list as an input to our decoder, with each element of the list being fed one by one to produce the desired 3D model sequence. In Figure 8 (a), we compare our results with direct linear interpolation between source and target 3D model output. Our results show that the interpolation in embedding space can avoid anatomical errors which usually occur in methods using direct linear interpolation. Further interpolation results can be found in the teaser image and the accompanying video.

## 5.4 Generation of Dynamic 3D models - Extrapolation

Our results show that the interpolation vector in the embedding space is capable of reasonably representing the motion in real world actions. To improve upon this idea, we exploit the learned interpolation vector in order to predict future movements. We show our results for extrapolation in Figure 8 (b). This figure shows that the learned interpolation vector between two 3D shapes contains meaningful information of movement in 3D. Further extrapolation results can be found in the teaser image and the accompanying video.

## 5.5 Timing

We finish our evaluations with our timing information. The closest work to our framework reports 10 minutes of production time for an amateur user to create 3D faces from 2D sketches [17]. In our system, on the other hand, it takes an amateur user 10 seconds of stick figure drawing and 1 second of algorithm processing to create 3D bodies from 2D sketches. There are two main reasons of this remarkable performance advantage of our framework: i) Human bodies can be naturally represented with easy-to-draw stick figures

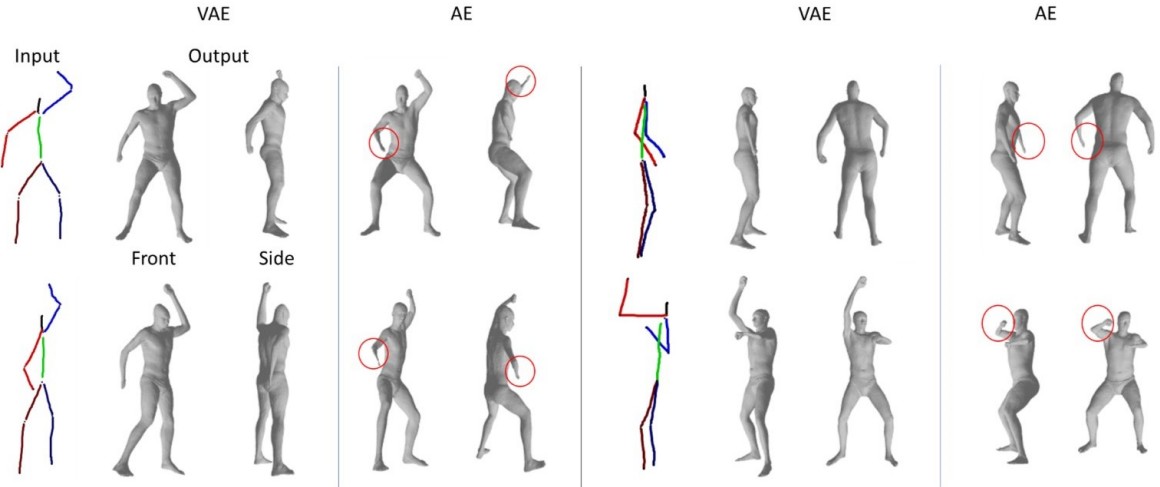

Figure 5: Generated 3D models in front and side views for given sketches using our VAE network and standard AE network. Flaws are highlighted with red circles.

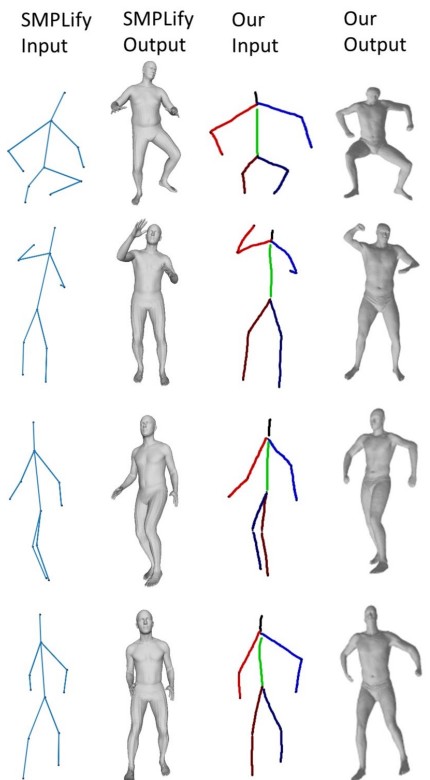

Figure 6: Qualitative comparison of our method (columns 3 and 4) with [20] (columns 1 and 2).

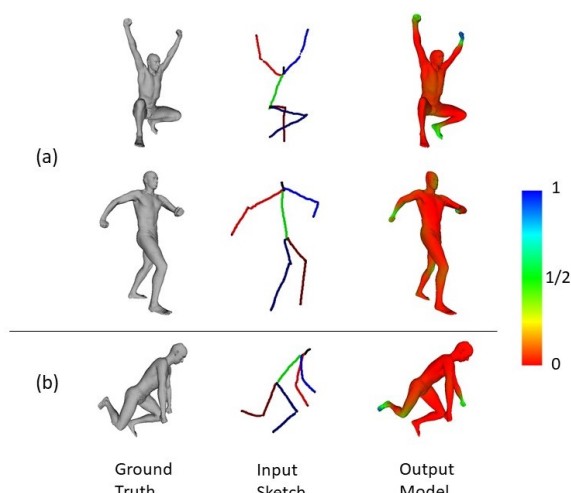

Figure 7: Generation results for (a) two sketches observed in training and (b) an unobserved sketch. The generated models (last column) are colored with respect to the distance map to the ground truth (first column). Distance values are normalized between 0 and 1.

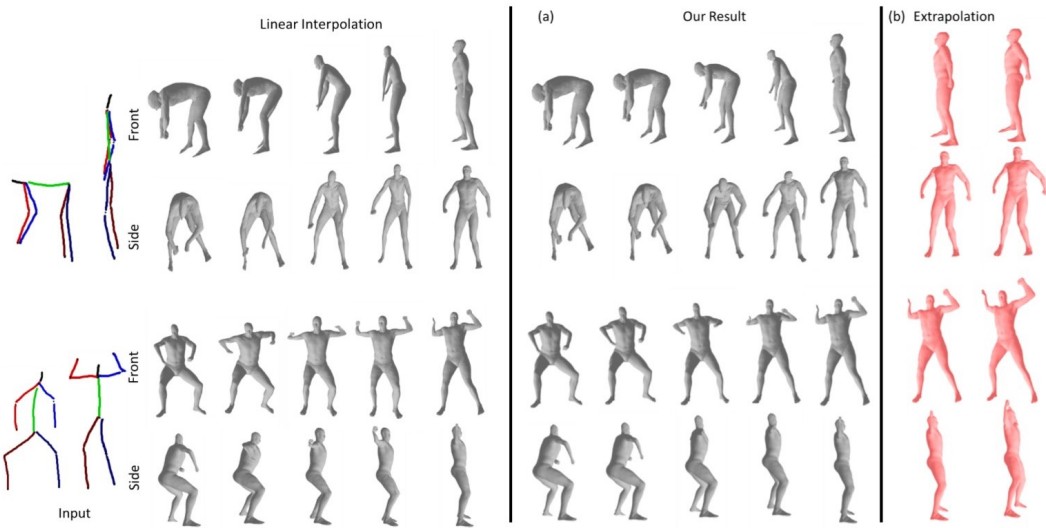

Figure 8: Qualitative comparison with linear interpolation. (a) Produced 3D model sequences for given sketches using our network are better than linear interpolation results. (b) Extrapolation results for 3D model sequences are given as red models.

whereas faces cannot. The simplicity and expressiveness make the learning easier and more efficient. ii) Our deep regression network is significantly less complex than the one employed in [17].

Our application runs on a PC with 8GB ram and i7 2.80GHz CPU. Training of our model is done on Tesla K40m GPU and took about 2 days (800 epochs total).

## 6 CONCLUSIONS

In this paper, we presented a deep learning based framework that is capable of generating 3D models from 2D stick figure sketches and producing dynamic 3D models between (interpolation) and beyond (extrapolation) two given stick figure sketches. Unlike existing methods, our method requires neither a statistical body shape model nor a rigged 3D character model. We demonstrated that our framework not only gives reasonable results on generation, but also compares favorably with existing approaches. We further supported our framework with a well-designed user interface to make it practical for a variety of applications.

## 7 LIMITATIONS

The proposed system has several limitations that are listed as follows:

- Training of our network is dependent on the existing 3D shapes in the dataset. Our network cannot learn vastly different shapes than existing ones: it produces incompatible 3D models with sketch inputs that are not closely represented in the dataset. For example, our network does not correctly capture the arm orientation for the right model in Figure 9.

- The system can only generate human shapes because of the content of the dataset.

- The system can produce articulated shapes. Although it can twist and bend human body in a reasonable way, it can not, for instance, stretch or resize a body part.

- The system benefits from the one-to-one correspondence information of the dataset. Thus, the quality of results depends on this information.

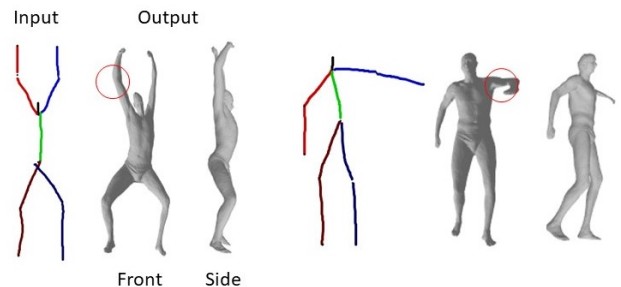

Figure 9: Failure cases of our framework. Our framework generates anatomically unnatural results if the input sketch has disproportionate body parts (left), or it is significantly different than the ones used during training (right).

- Since our network takes its input in a specific order, our user interface constrains users to sketch in that order. Users cannot sketch stick figures in an arbitrary order.

## 8 FUTURE WORK

Potential future work directions that align with our proposed system are described as follows. Human stick figures used in this paper can be generalized to any other shape using their skeletons. Consequently, an automated skeleton extraction algorithm would enable further training of our network, which in turn extends our solution to non-human objects. Voxelization of our input data would spare us from the one-to-one correspondence information requirement, which in turn would enable our interpolation scheme to morph from different object classes that do not often have this type of information, e.g., from cat to giraffe. Automatic one-to-one correspondence computation [22, 34, 36] can also be considered to avoid voxelization. Latent space can be exploited in a better manner in order to obtain a more sophisticated extrapolation algorithm than the basic one we introduced in this paper. New sketching cues can be designed and

incorporated into our network to be able to produce body types different than the one used during training, e.g., training with the fit SCAPE actor and production with a obese actress.

## ACKNOWLEDGMENTS

- This work was supported by TUBITAK under the project EEEAG-115E471.

- This work was supported by European Commission ERA-NET Program, The IMOTION project.

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
