# OpenReview forum: "Generation of 3D Human Models and Animations Using Simple Sketches"
_graphicsinterface.org/Graphics_Interface/2020/Conference — GI 2020_

### Official Review · AnonReviewer1 · 2020-02-04
**The method is alright, the results are not great -- but perhaps good enough**

**Rating:** 7
**Confidence:** 4

**Review:**

I have reviewed this paper earlier as a SIGGRAPH submission.

The paper presents a novel method based on deep learning to convert a rough stick figure, with joints drawn in a fixed pre-determined order, into a 3D human character in the intended shape. As a representation for the 3D shape, the authors chose a point cloud. The model has been trained on a tiny dataset (SCAPE), augmented with rotations. The architecture of the network is based on an autoencoder: a variational autoencoder learns the latent representation of the 3D point clouds for various poses, and a regular encoder learns a mapping from the 2D stick figure into the latent space. The paper is validated in various ways: doing the sketch-based posing of the character, pose interpolation, etc.

Since the SIGGRAPH submission, I was glad to see that the most important non-technical bits were done, such as missing references etc. Overall, the paper is in a good shape and easy to read. The results are not stellar, but certainly a worthy investigation. The model clearly has issues learning the latent space, because the dataset is so small, but it's still an interesting approach. Moreover, I think this area of sketch-based posing is largely underdeveloped, so I welcome contributions like this.

My only comment is really the lack of details in 4.4: how the models are visualized. Since the network outputs a point cloud, the authors combine it with the meshes from SCAPE, but I would like more details on how. Also, L797: "fat" -> "obese". I would also rethink the intro sentence "Despite the increasing number of talented artists" - not sure if that's what the authors intend to say really.

Other than that, I think the paper is ready for GI.

---

### Official Review · AnonReviewer3 · 2020-02-13
**A new but limited approach to sketch based modeling**

**Rating:** 6
**Confidence:** 2

**Review:**

The paper presents a system to obtain posed meshes of a human, based on input sketches of a stick figure. The method employs a neural network (VAE) to learn a latent space relating stick figures and meshes. The latent space can also be used to interpolate between poses.

Since I am not an expert on neural networks, I can not judge the network architecture, however, it seems to follow well-established design principles.

The paper is well written and good to follow. There are only a few grammar issues that can easily be resolved. The figures are a little hard to see. Maybe less intermediate steps (e.g. Figure 8) but larger images would be a good idea to make the differences between results more apparent.

The method relies on the SCAPE dataset and is limited to a specific character at a time given at different poses. The posed models have to be meshed consistently and specific landmark points defining a skeleton have been identified beforehand. I argue that this is enough information to easily find blend skinning weights and pose the character directly. In other words, the main problem has already been solved in this dataset. Of course one can argue that a stick figure is easier to draw and that automatic skinning has to be performed, however, compared to the 2 days of network training for just one specific model, this seems to be the better option. Moreover, the method shows severe artifacts, e.g. the head in Fig. 5, second row. Furthermore, if skinning weights are available, the matching could also be performed between 2d and 3d stick figures which appears to be much simpler.

That said, while quite limited, the method is still an innovative approach to a real problem and the paper could stimulate more research in this direction. Therefore I wouldn't argue against accepting the paper.

---

### Official Review · AnonReviewer2 · 2020-02-17
**The paper is weakly rejected or borderline.**

**Rating:** 5
**Confidence:** 3

**Review:**

This paper proposes a method to convert single-view 2D skeleton sketches to 3D human body poses based on VAE network architecture. The interface is easy to use. The paper was generally well-written and clear. But in my opinion, the results are not of high quality and the comparisons are not convincing enough. I intend to weakly reject the paper.
Some comments

- The input sketch/skeleton should be exactly the same when compared to [Kanazawa et al. 2018] in Section 5.2.
- The output of the sketch-based face modeling system [Han et al. 2017] is with fine details. So the production time is not comparable to this paper. And it is noticeable that the model inference based on their deep regression network takes 50ms, which is much faster than 1 second in this paper.
- The training dataset is quite small (only 72 poses) and the network produces incompatible 3D models with sketch inputs that are not closely represented in the dataset. And the interface provides no additional editing function to achieve the desired model when the direct output is not satisfactory. The application of this paper is relatively limited.

---

### Meta-Review · Area_Chair1 · 2020-02-19

**Recommendation:** Accept
**Confidence:** 3

**Metareview:**

The reviewers have agreed that the paper has results of borderline quality and limited applications (R1,R2,R3). However, the technique and the attempt itself are new rather interesting and might inspire new research (R1,R3).

I recommend accepting this paper, encouraging the authors to correct, if possible, the input skeletons in Section 5.2, and correct the comparison text with [Han et al. 2017] to address the concerns raised by R2. Similarly, I encourage the authors to correct the minor spelling mistakes and add the missing details, as suggested by R1, R3.

---

### Decision · Program_Chairs · 2020-02-21

Accept